# A new data set of nighttime chemical heating rates in the upper mesosphere and lower thermosphere derived from SCIAMACHY OH (9–6) emissions and SABER profiles

Xiaolin Wu<sup>1,2</sup>, Yajun Zhu<sup>1,3</sup>, Anne K. Smith<sup>4</sup>, Martin Kaufmann<sup>5</sup>, Guoying Jiang<sup>1,3,6</sup>, Shuai Liu<sup>1,2</sup>, and Jiyao Xu<sup>1,3</sup>

Correspondence: Yajun Zhu (y.zhu@swl.ac.cn)

Abstract. Chemical heating from exothermic reactions is a key component of the upper mesosphere–lower thermosphere (UMLT) energy budget, yet its quantification remains uncertain. We derive a new data set of heating rates at 22:00 local time for seven major reactions using Scanning Imaging Absorption Spectrometer for Atmospheric Chartography (SCIAMACHY) OH(9–6) limb emissions, collocated with Sounding of the Atmosphere using Broadband Emission Radiometry (SABER) temperature and ozone profiles. The retrieval assumes chemical equilibrium for ozone and  $HO_x$  and applies updated Einstein coefficients from HITRAN-2020. Consistent with earlier studies, the relative importance of individual reactions varies systematically with altitude: the hydrogen + ozone reaction (H +  $O_3$ ) provides the leading contribution below  $\sim$ 92 km, whereas three-body oxygen recombination (O + O + M) dominates above this level. Other reactions make a substantial contribution across much of the 80–96 km region, accounting for roughly one-third to one-half of the total heating above  $\sim$ 86 km. The derived latitude-altitude heating structures of the dominant reactions are significantly modulated by atmospheric tides. In the equatorial zone, these heating rates exhibit a pronounced semiannual cycle that tracks seasonal changes in temperature and key reactants. Relative to previous SCIAMACHY-based estimates, the updated data set yields lower heating rates from H +  $O_3$ . An uncertainty assessment indicates  $\sim$ 30% uncertainty for H +  $O_3$  and  $\sim$ 45–60% for O + O + M. These results refine and consolidate current understanding of chemical heating and its variability in the UMLT.

## 15 1 Introduction

The upper mesosphere–lower thermosphere (UMLT) region, spanning altitudes from approximately 80 to 120 km, is a critical coupling zone between the ionosphere and the lower atmosphere. Within this region, dynamics, energetics, and photochemistry are tightly coupled. Understanding its energy budget is essential for interpreting the thermal structure, circulation, chemical composition, as well as their interactions (Mlynczak and Solomon, 1993). The primary energy source in this region is solar

<sup>&</sup>lt;sup>1</sup>State Key Laboratory of Solar Activity and Space Weather, National Space Science Center, Chinese Academy of Sciences, Beijing, China

<sup>&</sup>lt;sup>2</sup>College of Earth and Planetary Sciences, University of Chinese Academy of Sciences, Beijing, China

<sup>&</sup>lt;sup>3</sup>Hainan National Field Science Observation and Research Observatory for Space Weather, Hainan, China

<sup>&</sup>lt;sup>4</sup>NSF National Center for Atmospheric Research, Boulder, Colorado, USA

<sup>&</sup>lt;sup>5</sup>Institute of Climate and Energy Systems - Stratosphere (ICE-4), Forschungszentrum Jülich, Jülich, Germany

<sup>&</sup>lt;sup>6</sup>School of Astronomy and Space Science, University of Chinese Academy of Sciences, Beijing, China

ultraviolet radiation absorbed by molecular oxygen and ozone. While part of the absorbed energy is immediately converted into heat, a significant portion is stored as chemical potential energy through photolysis (Mlynczak, 2000). The stored chemical energy is subsequently released as heat through various exothermic reactions, making chemical heating a critical component of the UMLT energy budget, particularly at night when direct solar heating is absent. Accurate quantification of chemical heating rates is crucial, as it not only deepens our understanding of the UMLT energy budget but also provides important insights into mesospheric inversion layers (Ramesh et al., 2017) and enables energetic constraints on the concentrations of key reactive species such as atomic oxygen (Mlynczak et al., 2013).

The importance of chemical heating was recognized early by Kellogg (1961), who proposed the hypothesis that energy released from atomic oxygen recombination could explain the anomalous warming of the polar winter mesosphere. However, accurately quantifying chemical heating remains challenging, primarily due to the difficulty in directly measuring the densities of key reactants, particularly atomic oxygen (O) and hydrogen (H). To overcome this challenge, comprehensive photochemical models were subsequently developed to provide systematic estimates of heating rates from the complex oxygen–hydrogen system (Crutzen, 1971; Hunt, 1972; Garcia and Solomon, 1985). Mlynczak and Solomon (1991) used the established two-dimensional model of Garcia and Solomon (1985) to assess the heating released by exothermic reactions involving odd-hydrogen species. Their results showed that these reactions were a major heat source in the upper mesosphere, sometimes becoming the dominant heating process.

In parallel with model development, observational constraints on the UMLT improved steadily. In particular, rocket-borne measurements of atomic oxygen, atmospheric density, and temperature have facilitated more reliable estimates of chemical heating rates. Based on high-latitude rocket data, Brasseur and Offermann (1986) found that the O + O + M reaction is a major source of heat in the UMLT, and that odd hydrogen species also play a significant role in the thermal budget. More recently, Grygalashvyly et al. (2024) retrieved nighttime chemical heating using rocket measurements over Andøya from three key reactions and found that vertically averaged chemical heating across the mesopause exceeds turbulent energy dissipation in narrow peaks a few hundred meters wide.

Satellite observations have provided additional insights into the spatiotemporal variability of the chemical heating rates. Riese et al. (1994) used atomic oxygen and hydrogen densities retrieved from the Solar Mesosphere Explorer (SME) hydroxyl (OH) airglow data (Thomas, 1990) to calculate the total chemical energy release from the main seven exothermic reactions. They found that the heating rates were significantly higher than those reported in previous studies. Kaufmann et al. (2008, 2013) estimated a peak chemical heating rate of about 10 K/day for the reaction between H and  $O_3$ , based on the Scanning Imaging Absorption Spectrometer for Atmospheric Chartography (SCIAMACHY) OH (9–6) spectral data. Further progress came with the Sounding of the Atmosphere using Broadband Emission Radiometry (SABER) instrument, where Mlynczak et al. (2013, 2018) retrieved O and H densities from the OH 2.0  $\mu$ m channel. The official SABER Level 2B heating-rate products, based on these retrievals, were subsequently used by Ramesh et al. (2014, 2017) to investigate the relationship between chemical heating and mesospheric inversion layers. More recently, Kulikov et al. (2024a) derived an independent data set of O, H, and chemical heating rates from SABER 2  $\mu$ m emissions, but used an OH emission model different from that of Mlynczak et al. (2013, 2018). These studies demonstrate that OH airglow emissions serve as an effective proxy for chemical heating rate

https://doi.org/10.5194/egusphere-2025-5463 Preprint. Discussion started: 14 November 2025

© Author(s) 2025. CC BY 4.0 License.

retrieval. Nevertheless, uncertainties in the OH emission mechanism remain a persistent challenge, contributing to substantial uncertainty in the estimated heating rates.

Recent updates to the key physical parameters in the OH emission model (Sharma et al., 2015; Kalogerakis et al., 2016), along with the updated Einstein coefficients from the HITRAN-2020 database (Gordon et al., 2022), provide an opportunity to reassess chemical heating rates in the UMLT region.

This study aims to derive a new data set of chemical heating rates for seven major exothermic reactions in the UMLT and to characterize their spatial and seasonal structures, based on SCIAMACHY OH(9–6) observations and SABER ozone and temperature data, using the latest physical parameters.

The structure of this paper is as follows. Section 2 introduces the data sets and the method for retrieving chemical heating rates. Section 3 presents the main results, including an uncertainty analysis, representative vertical profiles, and the spatial and seasonal variability of the heating rates. Section 4 compares the derived heating rates with other existing data sets. Section 5 summarizes the main findings of this study.

#### 2 Data and methods

#### 2.1 Data

The OH emission spectra used in this study were measured by the SCIAMACHY instrument (Bovensmann et al., 1999) onboard the European Environmental Satellite (Envisat), which operated from 2002 to 2012 in a sun-synchronous orbit. During nighttime limb observations, the instrument scanned tangent altitudes from approximately 73 to 148 km with a vertical step of about 3.3 km, covering latitudes typically from 30°S to 50°N and 0° to 80°N (Kaufmann et al., 2008). SCIAMACHY measured radiation across eight spectral channels from 220 to 2400 nm, with spectral resolution ranging from 0.2 to 1.5 nm depending on the channel (Gottwald et al., 2006). In this study, we use OH (9–6) limb emission spectra from channel 6, spanning 1377–1400 nm with a spectral resolution of 1.5 nm, recorded at 22:00 local time (LT). Channel 6 is well-calibrated and has been previously used to retrieve the densities of oxygen and hydrogen (Zhu and Kaufmann, 2018; Wu et al., 2025), as well as H + O<sub>3</sub> heating rates (Kaufmann et al., 2008, 2013).

Since SCIAMACHY lacked simultaneous nighttime measurements of ozone, temperature, and total density in the UMLT, these parameters were obtained from the SABER version 2.0 data set. SABER is a key payload on the Thermosphere, Ionosphere, Mesosphere, Energetics and Dynamics (TIMED) satellite and has been providing high-quality global measurements of mesosphere and lower thermosphere since 2002. The instrument is a limb-scanning multichannel radiometer that measures atmospheric temperature, airglow emissions, and constituent profiles (Russell et al., 1994; Mlynczak and Russell, 1995).

To improve the signal-to-noise ratio of the SCIAMACHY radiance data and to ensure sampling consistency between the two instruments, the data from both instruments were processed into monthly zonal medians. The median for each  $5^{\circ}$  latitude bin was calculated from the measurements within a spatiotemporal window of  $\pm 2.5^{\circ}$  in latitude and  $\pm 1$  hour at 22:00 LT. The mixing ratios of  $N_2$  and  $O_2$  were taken from the NRLMSIS 2.0 model (Emmert et al., 2021).

### 2.2 Hydroxyl night airglow model

In the UMLT, the odd-hydrogen family  $HO_x = H + OH + HO_2$  is tightly coupled by fast reactions. Because the partitioning among  $HO_x$  adjusts on seconds-to-minutes timescales at these altitudes, photochemical steady state is appropriate for OH and  $HO_2$  at night: their chemical production and loss rates approximately balance. The dominant reactions involve H, O,  $O_3$ , temperature, and the background species  $O_2$  and  $O_2$ , and are summarized in Table 1.

Hydroxyl (OH) forms primarily via the exothermic reaction  $H + O_3 \rightarrow OH(v) + O_2$ , which produces vibrationally excited OH (denoted OH(v)). OH is removed mainly by reactions with O and  $O_3$ .

For the excited vibrational manifold OH(v), we assume steady balance of level populations: for each v, production by  $O_3$  is balanced by radiative decay and collisional quenching by  $O_4$ , and  $O_4$ . Under optically thin conditions, the volume emission rate for a transition  $v \rightarrow v'$  depends linearly on the upper-state population and the corresponding Einstein A-coefficients.

This allows the retrieval of [O] and [H], where the square brackets denote the number density of the respective species, from OH (9–6) volume emission rates, following the method described by Zhu and Kaufmann (2018) and Wu et al. (2025):

100 VER<sub>(9-6)</sub> = 
$$\frac{f_9 \cdot k_1 \cdot [H] \cdot [O_3]}{A_9 + k_{O,9} \cdot [O] + k_{O_2,9} \cdot [O_2] + k_{N_2,9} \cdot [N_2]} \cdot A_{96} \cdot E_{96},$$
 (1)

In this expression,  $VER_{(9-6)}$  refers to the volume emission rate of the OH (9–6) airglow band measured by SCIAMACHY. The numerator represents the production rate of OH(v=9) via the exothermic reaction H + O<sub>3</sub>, where  $k_1$  is the rate constant for this reaction, and  $f_9$  is the branching ratio for the v=9 state, taken as 0.47 (Adler-Golden, 1997). The denominator represents the total loss of OH(v=9).  $k_{O_2,9}$  and  $k_{N_2,9}$  represent the quenching rate coefficients of OH(v=9) by O<sub>2</sub> and N<sub>2</sub>, respectively.  $k_{O,9}$  is the total loss rate of OH(v=9) with O by chemical and collisional quenching. These removal rate constants are consistent with those used in Zhu and Kaufmann (2018) and Wu et al. (2025). The parameter  $A_9$  refers to the total spontaneous emission rate for the v=9 state, and  $A_{96}$  corresponds to the sum of the Einstein coefficients for all ro-vibrational lines considered in the OH (9–6) band. Both values are taken from the HITRAN-2020 database (Gordon et al., 2022).  $E_{96}$  refers to the photon energy associated with the vibrational transition from v=9 to v=6.

At night in the UMLT ozone varies on timescales short enough that its chemical production and loss approximately balance (photochemical steady state). Photolysis is negligible, and the dominant terms are (i) three-body formation of ozone by  $O + O_2 + M \rightarrow O_3 + M$  and (ii) loss by reaction with atomic hydrogen and atomic oxygen,  $O + O_3 \rightarrow OO + O_3 \rightarrow OO + O_3$  are typically smaller at night in this altitude range and are neglected here for simplicity. This balance is most valid between  $\sim 80-96$  km at night. This results in the following expression:

115 
$$k_1 \cdot [H] \cdot [O_3] + k_7 \cdot [O] \cdot [O_3] = k_6 \cdot [O] \cdot [O_2] \cdot [M]$$
 (2)

 $k_6$  and  $k_7$  are the rate constants for the reactions O + O<sub>2</sub> + M and O + O<sub>3</sub>, respectively. M represents the background atmosphere. These reaction rates are taken from Burkholder et al. (2020).

Following the approach of Zhu and Kaufmann (2018) and Wu et al. (2025), we simultaneously solve for the densities of atomic oxygen and atomic hydrogen by combining the OH airglow model (Eq. 1) with the ozone chemical equilibrium equation

Table 1. Chemical reactions, rate constants, and enthalpy changes.

|       | Reaction                                                                       | Rate constant <sup>a</sup>                  | Enthalpy (kcal $mol^{-1}$ ) |
|-------|--------------------------------------------------------------------------------|---------------------------------------------|-----------------------------|
| (R1)  | $H + O_3 \rightarrow OH + O_2$                                                 | $k_1 = 1.4 \times 10^{-10} \exp(-470/T)$    | -76.90                      |
| (R2)  | $H + O_2 + M \rightarrow HO_2 + M$                                             | $k_2 = 5.3 \times 10^{-32} (298/T)^{1.8}$   | -49.10                      |
| (R3)  | $\mathrm{HO_2} + \mathrm{O} \rightarrow \mathrm{OH} + \mathrm{O_2}$            | $k_3 = 3.0 \times 10^{-11} \exp(200/T)$     | -53.27                      |
| (R4)  | $\mathrm{OH} + \mathrm{O} \rightarrow \mathrm{H} + \mathrm{O}_2$               | $k_4 = 1.8 \times 10^{-11} \exp(180/T)$     | -16.77                      |
| (R5)  | $\mathrm{O} + \mathrm{O} + \mathrm{M} \rightarrow \mathrm{O}_2 + \mathrm{M}$   | $k_5 = 4.7 \times 10^{-33} (298/T)^2$       | -119.40                     |
| (R6)  | $\mathrm{O} + \mathrm{O}_2 + \mathrm{M} \rightarrow \mathrm{O}_3 + \mathrm{M}$ | $k_6 = 6.1 \times 10^{-34} (298/T)^{2.4}$   | -25.47                      |
| (R7)  | $\mathrm{O} + \mathrm{O_3} \rightarrow 2\mathrm{O_2}$                          | $k_7 = 8.0 \times 10^{-12} \exp(-2060/T)$   | -93.65                      |
| (R8)  | $\rm H + \rm HO_2 \rightarrow 2OH$                                             | $k_8 = 7.2 \times 10^{-11}$                 |                             |
| (R9)  | $\mathrm{H} + \mathrm{HO_2} \rightarrow \mathrm{O_2} + \mathrm{H_2}$           | $k_9 = 6.9 \times 10^{-12}$                 |                             |
| (R10) | $\mathrm{H} + \mathrm{HO_2} \rightarrow \mathrm{O} + \mathrm{H_2O}$            | $k_{10} = 1.6 \times 10^{-12}$              |                             |
| (R11) | $\mathrm{OH} + \mathrm{O_3} \rightarrow \mathrm{O_2} + \mathrm{HO_2}$          | $k_{11} = 1.7 \times 10^{-12} \exp(-940/T)$ |                             |
| (R12) | $\mathrm{OH} + \mathrm{HO_2} \rightarrow \mathrm{H_2O} + \mathrm{O_2}$         | $k_{12} = 4.8 \times 10^{-11} \exp(250/T)$  |                             |

<sup>&</sup>lt;sup>a</sup> Units are cm<sup>3</sup> molecule<sup>-1</sup> s<sup>-1</sup> for two-body reactions and cm<sup>6</sup> molecule<sup>-2</sup> s<sup>-1</sup> for three-body reactions. Rate constants are taken from Burkholder et al. (2020).

(Eq. 2). The primary update in this study is the use of Einstein coefficients from the HITRAN-2020 database, replacing the HITRAN-2012 values used in earlier work.

In addition to [H] and [O], estimating chemical heating also requires the densities of OH and HO<sub>2</sub>, for which no direct observations are available. These densities are inferred assuming chemical equilibrium, with the lower boundary altitude ranging from 73 to 85 km depending on season and latitude (Kulikov et al., 2024b). The formulas for calculating the densities of OH and HO<sub>2</sub> are given below, with the corresponding reaction rate constants listed in Table 1:

$$[OH] = \frac{k_3 \cdot [HO_2] \cdot [O] + k_1 \cdot [H] \cdot [O_3] + 2k_8 \cdot [H] \cdot [HO_2]}{k_4 \cdot [O] + k_{11} \cdot [O_3] + k_{12} \cdot [HO_2]}$$
(3)

$$[HO_2] = \frac{k_2 \cdot [H] \cdot [O_2] \cdot [M] + k_{11} \cdot [OH] \cdot [O_3]}{k_3 \cdot [O] + (k_8 + k_9 + k_{10}) \cdot [H] + k_{12} \cdot [OH]}$$
(4)

# 2.3 Derivation of heating rates

The chemical heating in the mesopause region originates from the release of potential energy stored in chemical species produced by the solar photolysis of  $O_2$  and  $H_2O$ . This process generates the primary chemical families responsible for heating: the odd-oxygen family ( $O_x = O + O_3$ ), which acts as the main energy carrier, and the odd-hydrogen family ( $HO_x = H + OH + HO_2$ ), which drives highly efficient catalytic cycles. The stored energy is released back into heat through two main pathways: direct recombination of odd-oxygen (e.g., O + O + M) and catalytic reactions, among which the reaction between H

and  $O_3$  is particularly significant. To quantitatively evaluate the heat released by these processes, this study focuses on seven key exothermic reactions involving the  $O_x$  and  $HO_x$  families, as identified by Mlynczak and Solomon (1993). These reactions (R1–R7), along with their energy releases and rate constants from Burkholder et al. (2020), are detailed in Table 1.

It is important to note that the total energy released by these exothermic reactions is not always fully converted into atmospheric heating. A portion of the chemical energy can be transferred to the internal energy of the product molecules, which may then be radiated away as chemiluminescence before it can be thermalized through collisions. As a result, the chemical heating efficiency, defined as the ratio of energy converted to heat to the total energy released, can be less than one for certain reactions. The reaction of  $H + O_3$  is the most prominent example of this in the mesopause, as it produces vibrationally excited hydroxyl radicals, which subsequently radiate strongly in the Meinel bands. Based on a detailed evaluation, Mlynczak and Solomon (1993) recommended a heating efficiency of approximately 0.6 for this reaction. Smith et al. (2015) revisited the heating efficiency and found that the efficiency varies with atmospheric pressure, atomic oxygen concentration, and temperature, and that a value of 0.6 remains a good estimate for the global annual mean. Therefore, this study adopts an efficiency of 0.6 for the  $H + O_3$  reaction, and unit efficiency (1.0) for the other six reactions as recommended by Mlynczak and Solomon (1993).

Heating rates (dT/dt) for a given Reaction (Rx) can be calculated by the following formula:

$$\frac{dT}{dt} = \frac{2 \cdot \Delta E \cdot P_c \cdot \epsilon}{7k_B[M]} \tag{5}$$

Where  $\Delta E$  represents the exothermicity of the Reaction (Rx), and  $\epsilon$  is its heating efficiency. The term  $P_c$  is the rate of the reaction, calculated as the product of the reaction rate constant  $k_x$  and the number densities of the corresponding reactants. For example, for Reaction (R1) between H and O<sub>3</sub>,  $P_c$  is  $k_1 \cdot [H] \cdot [O_3]$ . While the number densities of reactants H, O, OH, and HO<sub>2</sub> required for calculating  $P_c$  are retrieved in this study, the remaining inputs such as O<sub>3</sub>, atmospheric density, and temperature are obtained from the SABER data set. The factor 2/7 originates from the relationship between the specific heat capacity of air at constant pressure and the gas constant.  $k_B$  is the Boltzmann constant and [M] is the total atmospheric number density.

#### 155 3 Results and discussion

165

## 3.1 Uncertainty analysis

In this section, we analyze the uncertainties in the heating rates for the dominant Reactions (R1)  $H + O_3$  and (R5) O + O + M. Uncertainty estimates are obtained by perturbing key input parameters and model coefficients including temperature,  $O_3$  density, collisional quenching rates, Einstein coefficients, and heating efficiency, and then quantifying their respective impact on the calculated heating rates.

The SABER kinetic temperature uncertainty is taken as 2 K at 80 km and increases to 7–8 K at 96 km (Remsberg et al., 2008). The uncertainty in  $O_3$  density is taken to be 20% (Smith et al., 2013). For the collisional rate coefficients, we adopt uncertainty ranges of  $(2.3\pm1)\times10^{-10}~{\rm cm^3\,s^{-1}}$  for  $k_{O,9}$ ,  $(2.2\pm0.6)\times10^{-11}~{\rm cm^3\,s^{-1}}$  for  $k_{O_2,9}$ , and  $(7\pm2)\times10^{-13}~{\rm cm^3\,s^{-1}}$  for  $k_{N_2,9}$  (Kalogerakis et al., 2011; Zhu and Kaufmann, 2018). HITRAN-2020 does not provide explicit uncertainties for Einstein A-coefficients, as they are assumed to share the same uncertainties as line intensities (Gordon et al., 2022). For the

OH(9–6) lines used here, the HITRAN uncertainty code indicates 10–20%; we therefore adopt 20% as a representative value. For heating efficiency, we assume  $0.6 \pm 0.1$  for Reaction (R1) (Smith et al., 2015; Grygalashvyly et al., 2024).

For the heating rates of Reaction (R1), our analysis shows that the uncertainty in the collisional quenching rate  $k_{\rm O_2,9}$  has the largest impact, causing a perturbation of approximately 20%. Heating efficiency and Einstein coefficients are also major sources of uncertainty, causing approximately 17% and 15% perturbations to the results, respectively. The influence of other parameters is smaller. Uncertainties in quenching rates  $k_{\rm O,9}$  and temperature each lead to a perturbation of approximately 10% in the calculated heating rate. The remaining factors, including quenching rates  $k_{\rm N_2,9}$  and  $k_{\rm N_2,9}$  has estimated to be approximately 30% at 80–96 km, dominated by the uncertainties in  $k_{\rm N_2,9}$ , heating efficiency, and the Einstein coefficients.

For Reaction (R5), temperature is the dominant source of uncertainty, inducing a 20–60% perturbation in the primary heating region above 90 km. Other major contributors include the Einstein coefficients and the collisional quenching rate  $k_{\rm O,9}$ , inducing uncertainties of approximately 30% and 20–30%, respectively. The impact of  $k_{\rm O_2,9}$  is moderate, introducing a 15–30% variation, whereas the effects of  $\rm O_3$  density and  $k_{\rm N_2,9}$  are smaller, at around 10% and 5%, respectively. The total RSS uncertainty for Reaction (R5) is estimated to be 45–60% at 80–96 km, primarily driven by temperature, the Einstein coefficients and  $k_{\rm O.9}$ .

Finally, the total RSS uncertainty for the combined chemical heating from all seven Reactions (R1–R7) is estimated to be 25–55% between 80 and 96 km, with a peak value of 55% at 96 km. The dominant sources of uncertainty vary with altitude. Above 90 km, temperature is the largest contributor, introducing a perturbation of about 40% at 96 km. The second-largest contributor in this region is the quenching rate  $k_{\rm O,9}$ , causing a perturbation of approximately 30% at 96 km. Below 90 km,  $k_{\rm O_2,9}$  is the dominant factor, introducing a perturbation of about 20%. Additionally, the uncertainty in the Einstein coefficients remains non-negligible across 80–96 km, causing a perturbation of approximately 15–20%.

## 3.2 Heating rate profiles

Figure 1 displays the chemical heating rate profiles for the seven chemical reactions and their total, calculated using Eq. (5) and averaged over June–August 2007, for the 20– $40^{\circ}$ S (left),  $10^{\circ}$ S– $10^{\circ}$ N (middle), and 20– $40^{\circ}$ N (right) latitude bands. The results indicate that Reaction (R1) H + O<sub>3</sub> is the primary chemical heating source below 92 km, with heating rates peaking at 4–5 K near 90 km and a maximum relative contribution of 45% to the total heating at 86 km. Above 92 km, the Reaction (R5) O + O + M becomes the main contributor, peaking at 3–4 K around 93–96 km, and accounting for up to 50% of the total heating at 96 km. The contributions from Reaction (R2) and (R3) are minimal, with maximum values of only 0.5–1 K near 83 km. Their similar vertical profiles are expected, as the product of Reaction (R2) (HO<sub>2</sub>) serves as a reactant in Reaction (R3). The heating rates from Reaction (R4) peak at around 90 km with a maximum of 1.7 K, contributing about 15% to the total heating below 90 km. Reaction (R6) reaches a peak heating of about 2.6 K near 90 km. The altitude profile of Reaction (R6) closely resembles that of Reaction (R1), as Reaction (R6) produces O<sub>3</sub>, which acts as a key reactant in Reaction (R1). For Reaction (R7), the maximum heating rate is less than 1 K around 93 km, with negligible contribution below 85 km. Overall, the total chemical heating rate peaks around 11 K at 90–93 km and decreases away from this altitude range.

210

**Figure 1.** Chemical heating rate profiles (K/day) from seven exothermic Reactions (R1–R7) and their total sum, averaged over June–August 2007, for 20–40°S, 10°S–10°N, and 20–40°N latitude bands between 80 and 100 km at 22:00 local time. Horizontal dotted lines denote the estimated uncertainties for Reactions (R1), (R5), and the total chemical heating.

The analysis above reveals that Reaction (R1) H + O<sub>3</sub> and Reaction (R5) O + O + M are the two dominant contributors to chemical heating. Reaction (R1) serves as the main heating source below approximately 92 km, with its rate governed by the local densities of H and O<sub>3</sub>, as well as the ambient temperature, whereas Reaction (R5) dominates above this altitude, with its rate primarily driven by the atomic oxygen concentration. Further calculations show that the total heating from the odd-hydrogen reactions (R1–R4) decreases with altitude, while that from the odd-oxygen reactions (R5–R7) increases, and the two contributions become comparable near 92 km.

This vertical structure, characterized by a crossover altitude around 92 km, is a fundamental feature of the nighttime UMLT energy budget and is consistent with previous research. Early modeling studies showed that heating from odd-hydrogen chemistry can exceed that from odd-oxygen reactions between 70 and 90 km (Mlynczak and Solomon, 1991). SME observations demonstrated that odd-oxygen reactions dominate near 93 km, while at lower altitudes the influence of atomic oxygen decreases rapidly, giving way to odd-hydrogen reactions (Riese et al., 1994). More recent analyses using modern data sets further support this view: SABER data presented in Ramesh et al. (2017) revealed a similar crossover structure in tropical latitudes, and the WADIS-2 rocket campaign reported a comparable crossover altitude of ~95 km at 69°N (Grygalashvyly et al., 2024). Thus, our findings, derived from SCIAMACHY OH limb emissions collocated with SABER atmospheric profiles, provide a robust, independent confirmation of this UMLT energy budget feature.

Figure 2. Latitude–altitude distribution of chemical heating rates for Reaction (R1)  $H + O_3$  in 2007, derived from SCIAMACHY OH (9–6) emission data and SABER atmospheric profiles. The numbers indicate the month of the year.

Although Reactions (R1) and (R5) are the primary sources of heating, the combined contribution from other chemical reactions is also non-negligible. Above 86 km, these other reactions contribute around 30–50% to the total chemical heating, while below 86 km, their contribution exceeds 50%.

### 3.3 Latitude-altitude distribution

The latitude–altitude distribution of the chemical heating rate from Reaction (R1) for each month of 2007 is illustrated in Fig. 2. The heating peak occurs around 85 to 90 km, with peak rates of 4–8 K/day, which vary with season and latitude. Heating rates are generally higher in spring and autumn, particularly at the equator and in the mid-latitudes. By averaging all low- to mid-latitude profiles from 2003 to 2011, we find that the heating peak for Reaction (R1) is approximately 4–5 K/day at 85–90 km.

Figure 3 shows the latitude–altitude distribution of chemical heating rates for Reaction (R5) O + O + M, in 2007. Reaction (R5) primarily produces heating above 90 km, with peak values near 96 km, where atomic oxygen reaches its maximum concentration. Its peak magnitude varies substantially, ranging from 4 to 18 K/day depending on season and latitude. In terms

Figure 3. Latitude–altitude distribution of chemical heating rates for Reaction (R5) O + O + M in 2007, derived from SCIAMACHY OH (9–6) emission data and SABER atmospheric profiles. The numbers indicate the month of the year.

of latitude, the largest heating rates are typically centered around 30°N, reaching a maximum of 18 K/day in October. However, in April, substantial heating also occurs in the equatorial region, with rates of about 10 K/day and heating extending downward to around 85 km. Based on calculations from 2003 to 2011, the averaged peak heating rate for Reaction (R5) is around 5 K/day at 92–96 km.

The distribution of total heating rates from the seven exothermic reactions in 2007 is presented in Fig. 4. The peak of the total chemical heating occurs between 85 and 96 km, with magnitudes ranging from 10 to 38 K/day. Both the peak altitude and intensity vary with season and latitude. The peak altitude around 30°N is higher than that at the equator. This is because at 30°N, the contribution from Reaction (R5), which peaks near 96 km, becomes more significant compared to Reaction (R1), effectively increasing the altitude of the total heating maximum. Seasonally, the heating rates are lower in summer compared to other seasons. Based on the average of low- to mid-latitude profiles from 2003 to 2011, the peak of the total heating rate is approximately 14 K/day at 90–93 km.

It is important to note that the results in Figs. 2–4 represent derived heating rates at 22:00 LT, and that their structures are subject to modulation by atmospheric tides. To assess this influence, we present the temperature perturbation at 22:00 LT in Fig. 5. The perturbation, obtained by subtracting the diurnally averaged temperature (over a 60-day window) from the

**Figure 4.** Latitude–altitude distribution of total heating rates of seven exothermic Reactions (R1–R7) in 2007, derived from SCIAMACHY OH (9–6) emission data and SABER atmospheric profiles. The numbers indicate the month of the year.

monthly mean temperature at 22:00 LT, represents the thermal signature of tidal activity. The figure reveals significant positive perturbations in the equatorial region at 80–90 km and in the subtropical regions above 90 km. This spatial pattern corresponds to the enhanced  $H + O_3$  heating near the equator shown in Fig. 2 and the O + O + M heating enhancement near 96 km in the subtropics shown in Fig. 3.

This correspondence suggests that tidal activity is a key factor influencing the heating patterns in the UMLT region. In particular, tidal temperature perturbations can directly modulate chemical heating through the temperature-dependent reaction rate coefficients. Moreover, vertical motions associated with the migrating diurnal tide drive downward transport of O-rich air, leading to enhanced atomic oxygen concentrations in this region (Smith et al., 2010; Jones et al., 2014). As shown in Zhu and Kaufmann (2018), the atomic oxygen distribution exhibits a latitude–altitude structure very similar to the tidal pattern in Fig. 5, with a pronounced peak near 96 km in the subtropics. Since the R5 heating rate is proportional to the square of [O], the enhanced [O] substantially amplifies the heating. Furthermore, this increase in atomic oxygen also promotes ozone production through the  $O + O_2 + M$  reaction, which in turn enhances the  $O + O_3$  heating rate and contributes to the patterns shown in Fig. 2.

**Figure 5.** Latitude-altitude distribution of the monthly mean SABER temperature perturbation at 22:00 LT for 2007. The perturbation, calculated as the monthly mean temperature at 22:00 LT minus the diurnally averaged temperature (over a 60-day window)

### 3.4 Seasonal variation

265

In this section, we examine the seasonal variations of the heating rates for the primary chemical Reactions (R1) and (R5), as well as for the total chemical heating. Fig. 6 shows the time–altitude distributions from 2003 to 2011 for Reaction (R1) heating rates, temperature, and O<sub>3</sub> density in the equatorial region (10°S–10°N). The mean chemical heating rate of Reaction (R1) in this region is approximately 5–6 K/day at 85–90 km. A clear semiannual oscillation (SAO) is evident for Reaction (R1), with maxima occurring near the equinoxes and stronger amplitudes in spring. As seen in Fig. 6(b) and Fig. 6(c), this seasonal variation strongly correlates with temporal changes in both temperature and O<sub>3</sub> density, which is driven by the semiannual cycle of the migrating diurnal tide. This relationship is expected, as higher temperatures and increased O<sub>3</sub> concentrations enhance the reaction rate, resulting in greater chemical heating.

Figure 7 shows the seasonal variations in heating rates of Reaction (R5) and the atomic oxygen densities in the equatorial region (10°S–10°N). The mean chemical heating rate of Reaction (R5) in this region is approximately 3–4 K/day between 90 and 96 km. Similar to Reaction (R1), the heating rates for Reaction (R5) exhibit a semiannual cycle, with peaks occurring

Figure 6. Time-altitude distributions from 2003 to 2011 in the equatorial region ( $10^{\circ}$  S- $10^{\circ}$  N) for (a) Reaction (R1) heating rates (K day<sup>-1</sup>), (b) temperature (K), and (c) O<sub>3</sub> density (cm<sup>-3</sup>) over 80–96 km. White regions indicate data gaps.

during the spring and autumn equinoxes, and notably stronger during spring. These variations closely follow the seasonal cycle of atomic oxygen, which is strongly modulated by downward transport driven by the migrating diurnal tide.

Figure 8 shows the seasonal variations in the total heating rates from Reactions (R1–R7) in the equatorial region (10°S–10°N). The mean total heating rate is approximately 12–15 K/day between 85 and 93 km. The heating rates exhibit a clear semiannual cycle, with enhanced heating during the equinox seasons, especially in spring. The total chemical heating is primarily dominated by Reactions (R1) and (R5), and its vertical and temporal structure is largely controlled by the combined influences of O<sub>3</sub>, O, and temperature.

#### 4 Comparisons with other datasets

270

Kaufmann et al. (2008) were the first to use SCIAMACHY OH (9–6) emissions to retrieve heating rates for the H + O<sub>3</sub> reaction, based on Einstein coefficients from HITRAN-2004. However, the HITRAN-2004 data set was later found to contain a programming error and incorrect parity assignments (e and f) in the pure rotational bands (Rothman et al., 2009). Subsequently, Kaufmann et al. (2013) updated their retrievals using corrected Einstein coefficients and re-estimated the associated chemical heating rates. Their results showed that the peak heating rates for the H + O<sub>3</sub> reaction in the equatorial region are approximately

Figure 7. Time-altitude distributions from 2003 to 2011 in the equatorial region  $(10^{\circ}\text{S}-10^{\circ}\text{N})$  for (a) Reaction (R5) heating rates (K day<sup>-1</sup>) and (b) O density (cm<sup>-3</sup>) over 80–96 km. White regions indicate data gaps.

**Figure 8.** Time–altitude distributions from 2003 to 2011 in the equatorial region  $(10^{\circ}\text{S}-10^{\circ}\text{N})$  for total heating rates  $(\text{K day}^{-1})$  from Reactions (R1–R7) over 80–96 km. White regions indicate data gaps.

8–10 K/day, whereas our results yield lower values of approximately 5–6 K/day. This substantial discrepancy mainly arises from their use of a higher quenching rate  $k_{O_2,9}$  and a lower nascent fraction  $f_9$  relative to the values used here, as discussed in detail by Wu et al. (2025). Therefore, our results, based on the latest physical parameters, represent an update to those reported by Kaufmann et al. (2013).

**Figure 9.** Latitude—altitude distribution of percentage differences for 2008 between heating rates for Reaction (R1) (top panel) and Reaction (R5) (bottom panel) derived from SABER and those derived from SCIAMACHY in this work. The SABER-based heating rates were calculated using the atomic oxygen and hydrogen densities reported by Mlynczak et al. (2018).

We also compare our heating rates with those calculated using atomic oxygen and hydrogen profiles retrieved from SABER observations. Mlynczak et al. (2013) derived O and H densities from SABER 2  $\mu$ m emissions, but these were later revised using updated model coefficients (Mlynczak et al., 2018). In this study, we use the updated O and H profiles from Mlynczak et al. (2018) to compute heating rates for Reactions (R1) and (R5) and compare them to our SCIAMACHY-derived results. The background atmospheric parameters used in the calculations, including temperature and O<sub>3</sub>, are taken from SABER.

Figure 9 shows the percentage differences between the chemical heating rates derived from SCIAMACHY and those derived from SABER for Reactions (R1) and (R5). For the H + O<sub>3</sub> reaction (R1, top panel), the two data sets exhibit a distinct vertically dependent discrepancy. Above 92 km, the SABER-derived rates are generally higher than our results, with the difference reaching approximately 35% at 96 km. In contrast, below 92 km, the SABER rates are 20–45% lower. A similar vertical pattern is observed for the O + O + M reaction (R5, bottom panel): SABER-derived rates exceed the SCIAMACHY estimates by 10–25% above 95 km, while below 90 km, they are 40–80% lower. These discrepancies are expected, as they directly reflect the known systematic differences in atomic oxygen and hydrogen densities retrieved from SCIAMACHY and SABER, which have been analyzed in detail by Zhu and Kaufmann (2018) and Wu et al. (2025), respectively.

#### 5 Conclusions

In this study, we present a new data set of chemical heating rates for seven primary exothermic reactions in the UMLT region, derived from SCIAMACHY OH (9–6) limb emissions and collocated SABER temperature and ozone profiles. The retrieval was based on the assumption of chemical equilibrium for ozone and  $HO_x$  species and uses Einstein coefficients from the HITRAN-2020 database. The observations are for 22:00 local time. Because of the influence of the diurnal tide on chemical and physical processes, they may differ somewhat from nighttime average heating rates.

Our results indicate that the reactions  $H + O_3$  and O + O + M are the dominant heating sources below and above  $\sim$ 92 km, respectively. In low-to-mid latitudes, the heating rate of  $H + O_3$  peaks on average at 4–5 K/day between 85 and 90 km, while that for O + O + M peaks on average at 5 K/day between 92 and 96 km. Besides these two main reactions, other exothermic reactions also make non-negligible contributions, accounting for 30–50% of the total heating above 86 km. The total nighttime heating peaks around 90–93 km at  $\sim$ 14 K/day. Additionally, in the equatorial region, the heating rates of  $H + O_3$ , O + O + M, and the total of all seven reactions exhibit a clear semiannual variation, with stronger peaks around the spring equinox, mainly driven by seasonal changes in temperature and key reactants ( $O_3$  and O).

Comparisons with other heating rate data sets show substantial differences. Our results provide a significantly lower estimate for the  $H + O_3$  heating rate, at approximately half the value of that reported in previous SCIAMACHY-based work (Kaufmann et al., 2013), representing an update based on the latest physical parameters. Furthermore, a comparison with SABER-derived heating rates reveals significant, altitude-dependent discrepancies for the dominant reactions. These differences are shown to be a direct consequence of the known systematic differences in the O and H profiles retrieved from the two satellite instruments.

It is worth noting that our analysis also reveals considerable uncertainties in deriving heating rates from OH airglow. The uncertainty is estimated to be approximately 30% for R1 and 45–60% for R5. The largest sources of the uncertainty are

325

330

background temperature, collisional rate coefficients, and the Einstein coefficients. These retrieval uncertainties, along with the large differences in heating rates derived from SCIAMACHY and SABER, indicate that accurately quantifying the UMLT chemical heating budget remains a persistent challenge. Furthermore, our results provide an important observational constraint on UMLT heating that will be valuable for atmospheric modeling and for understanding the mesopause energy budget.

320 Data availability. The SCIAMACHY Level 1b version 10 data used in this study are available at https://hm-atmos-ds.eo.esa.int/oads/access/collection/Envisat\_SCIAMACHY\_Level\_1b\_SCI\_\_\_\_\_1P/tree. SABER version 2.0 data are available at https://saber.gats-inc.com/browse\_data.php.

Author contributions. XW processed the data, performed the analysis, and drafted the manuscript. YZ initiated the topic, processed the initial data, and provided supervision. MK, AKS, and JX assisted in the review and editing, and provided suggestions. GJ and SL provided suggestions through helpful discussion.

Competing interests. The authors declare that they have no conflict of interest.

Acknowledgements. This work is supported by the Project of Stable Support for Youth Team in Basic Research Field, CAS (YSBR-018), the National Natural Science Foundation of China (42174212), and the Chinese Meridian Project. This material is also based upon work supported by the NSF National Center for Atmospheric Research, which is a major facility sponsored by the U. S. National Science Foundation under Cooperative Agreement No. 1852977.

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
