# Peer review of "A new data set of nighttime chemical heating rates in the upper mesosphere and lower thermosphere derived from SCIAMACHY OH (9–6) emissions and SABER profiles"

_EGUsphere, 2025_

## Referee Comment (RC1)

**Review of the paper "A new data set of nighttime chemical heating rates in the upper mesosphere and lower thermosphere derived from SCIAMACHY OH (9–6) emissions and SABER profiles"**

As is known, the chemical heat due to exothermic reactions between $HO_x$-$O_x$ species is one of the major sources in the mesopause energy balance. In particular, this source can exceed the heat from the turbulent energy dissipation, see, e.g., Grygalashvyly et al. (Earth Planets Space, 2024). Therefore, the subject is important and within the scope of ACP. At these altitudes, the local (in time and space) chemical heating rate is determined by the local concentrations of O, H, $O_3$, OH, $HO_2$, and air, as well as the temperature. Most of these variables are difficult to measure directly, but can be evaluated, for example, from satellite data according to photochemical and airglow models that link measured and derived variables. At this time, only data of the SABER/TIMED satellite campaign make it possible to calculate the chemical heating rate (CHR) both locally and globally, see https://saber.gats-inc.com. This website presents long-term spatiotemporal datasets of CHR for each major exothermic reaction with high space-time resolution. The used approach was two-step. First, O and H profiles were derived from local SABER data (profiles of temperature, air concentration, $O_3$, and total OH* emission rate due to 9→7 and 8→6 transitions) combining three algebraic equations that follow from equilibrium assumptions for $O_3$ and exited OH in the $v$=8 and $v$=9 states (Mlynczak et al., JGR, 2013, 2014). Second, the obtained O and H profiles were used for retrieving OH and $HO_2$ profiles from their equilibrium equations. Note there was much discussion of the SABER O and H quality retrieval (see, e.g., Kaufmann et al., GRL, 2014) in the context of the OH* excitation and relaxation parameters. Recently, Mlynczak et al. (GRL, 2018) improved the O and H retrieval and therefore CHR data quality (Zhu and Kaufmann, GRL, 2018.), but the questions, for example, with OH* model uncertainties remain. Thus, new CHR data of higher quality than those obtained from the SABER data are certainly very important and welcome.

In the reviewed paper, the nighttime CHR is derived from OH*(9→6) emission measured by SCIAMACHY/ENVISAT with the use of following updates in the OH* model: new Einstein coefficients from HITRAN-2020 and new rates of the reaction OH($v$)+O($^3$P) according to Sharma et al., (GRL, 2015) and Kalogerakis et al. (GRL, 2016). The retrieval techniques is similar to above mentioned. First, nighttime O and H profiles are derived with the use of two algebraic equations following from equilibrium assumptions for nighttime $O_3$ and OH($v$=9). Second, nighttime OH and $HO_2$ are obtained from their equilibrium conditions and then the CHR calculation is performed. Note the use of a simpler (more reliable) OH* model with updated parameters allows us to hope for an improvement in the quality of CHR retrieval. However, SCIAMACHY did not measure temperature,

total density and ozone profiles and therefore the authors were forced to use SABER data. During the ENVISAT operation time, SCIAMACHY and SABER had different flight geometry, making it quite difficult to collocate the data measured by two devices in both time and space, i.e., find a sufficient number of pairs of SCIAMACHY and SABER measurements corresponding to close values of time, latitude and longitude simultaneously. Therefore, the authors, first, average the initial measured data near local time 22 hours over a month and only then start the retrieval procedures. Note all mentioned algebraic equations are nonlinear. Evidently, there is a question about the correctness of the approach, but the paper does not provide an analysis of possible errors caused by such collocation of data measured by two independent instruments. The authors compared new CHR data with SABER data and revealed essential systematic differences. At the same time, the article does not provide convincing reasons to believe that the quality of the new CHR data is better than SABER CHR data, which exceeds significantly the new data in spatiotemporal resolution. In addition, there are no restrictions on obtaining CHR data at the altitudes of 80-85 km, where, according to Kulikov et al. (ACP, 2023, 2024), the equilibrium conditions of nighttime $O_3$, OH, and $HO_2$ may be disturbed.

Based on this short analysis, I find it difficult to assess the scientific importance and value of the presented results.

**Specific comments**

1. Lines 110-115. The authors claim that the chemical production and loss of nighttime ozone are in equilibrium within the range 80–96 km and further apply this assumption to the chemical heating rate retrieval in this altitude range. This contradicts the results of Kulikov et al. (JGR 2018, ACP 2023) showing that the lower boundary of the ozone balance depends on the season and latitude and can be located at an altitude of several kilometers above 80 km. In particular, Kulikov et al. (ACP 2023) presented the spatiotemporal evolution of this boundary retrieved from the SABER/TIMED data in 2002–2021. Below, one can see the time evolution of monthly mean altitude of this boundary at different latitudes reprinted from Figure 11 of the paper.

[Figure]

In addition, Kulikov et al. (JGR, 2018) showed by a 3D simulation (see Fig. 5 there) that the difference between dynamically calculated (i.e., by differential equation) and equilibrium $O_3$ concentrations below the mentioned boundary might reach up to 100%. It means the disturbance of ozone equilibrium approximation may be a major source of errors in the retrieval procedure.

2. Lines 122-127. The authors noted here: "These densities are inferred assuming chemical equilibrium, with the lower boundary altitude ranging from 73 to 85 km depending on season and

latitude (Kulikov et al., 2024b)." Nevertheless, as one can see from text and Figures of this paper, the chemical heating rate is calculated in the altitude range of 80-96 km, regardless of season and latitude.

*References*

Grygalashvyly, M., Strelnikov, B., Strelnikova, I. et al. Chemical heat derived from rocket-borne WADIS-2 experiment. Earth Planets Space **76**, 180 (2024). https://doi.org/10.1186/s40623-024-02129-x

Kalogerakis, K. S., Matsiev, D., Sharma, R. D., and Wintersteiner, P. P.: Resolving the mesospheric nighttime 4.3 μm 375 emission puzzle: Laboratory demonstration of new mechanism for OH($v$) relaxation, Geophys. Res. Lett., 43, 8835−8843, https://doi.org/10.1002/2016GL069645, 2016.

Kaufmann, M., Zhu, Y., Ern, M., & Riese, M. (2014). Global distribution of atomic oxygen in the mesopause region as derived from SCIAMACHY O(1S) green line measurements. Geophysical Research Letters, 41, 6274–6280. https://doi.org/10.1002/2014GL060574

Kulikov, M. Y., Belikovich, M. V., Grygalashvyly, M., Sonnemann, G. R., Ermakova, T. S., Nechaev, A. A., and Feigin, A. M.: Nighttime ozone chemical equilibrium in the mesopause region. J. Geophys. Res., 123, 3228–3242, https://doi.org/10.1002/2017JD026717, 2018.

Kulikov, M. Yu., Belikovich, M. V., Chubarov, A. G., Dementyeva, S. O., and Feigin, A. M.: Boundary of nighttime ozone chemical equilibrium in the mesopause region: long-term evolution determined using 20-year satellite observations, Atmos. Chem. Phys., 23, 14593–14608, https://doi.org/10.5194/acp-23-14593-2023, 2023.

Kulikov, M. Y., Belikovich, M. V., Chubarov, A. G., Dementyeva, S. O., and Feigin, A. M.: Technical note: Nighttime OH and HO2 chemical equilibria in the mesosphere−lower thermosphere, Atmos. Chem. Phys., 24, 10 965−10 983, https://doi.org/10.5194/acp-24-10965-2024, 2024.

Mlynczak, M. G., Hunt, L. A., Mast, J. C., Marshall, B. T., Russell III, J. M., Smith, A. K., Siskind, D. E., Yee, J.-H., Mertens, C. J., Martin-Torres, F. J., Thompson, R. E., Drob, D. P., and Gordley, L. L.: Atomic oxygen in the mesosphere and lower thermosphere derived from SABER: Algorithm theoretical basis and measurement uncertainty, J. Geophys. Res., 118, 5724–5735, https://doi.org/10.1002/jgrd.50401, 2013.

Mlynczak, M. G., Hunt, L. A. Marshall, B. T. Mertens, C. J. Marsh, D. R. Smith, A. K. Russell, J. M. Siskind D. E., and Gordley L. L.: Atomic hydrogen in the mesopause region derived from SABER: Algorithm theoretical basis, measurement uncertainty, and results, J. Geophys. Res., 119, 3516–3526, https://doi.org/10.1002/2013JD021263, 2014.

Mlynczak, M. G., Hunt, L. A., Russell, J. M. III, and Marshall, B. T.: Updated SABER night atomic oxygen and implications for SABER ozone and atomic hydrogen, Geophys. Res. Lett., 45, 5735–5741, https://doi.org/10.1029/2018GL077377, 2018.

Sharma, R. D., Wintersteiner, P. P., and Kalogerakis, K. S.: A new mechanism for OH vibrational relaxation leading to enhanced $CO_2$ emissions in the nocturnal mesosphere, Geophys. Res. Lett., 42, 4639–4647, https://doi.org/10.1002/2015GL063724, 2015.

Zhu, Y. and Kaufmann, M.: Atomic oxygen abundance retrieved from SCIAMACHY hydroxyl nightglow measurements, Geophys. Res. 440 Lett., 45, 9314–9322, https://doi.org/10.1029/2018GL079259, 2018.